# An Injectable Click-Crosslinked Hydrogel that Prolongs Dexamethasone Release from Dexamethasone-Loaded Microspheres

**DOI:** 10.3390/pharmaceutics11090438

**Published:** 2019-09-01

**Authors:** Ji Yeon Heo, Jung Hyun Noh, Seung Hun Park, Yun Bae Ji, Hyeon Jin Ju, Da Yeon Kim, Bong Lee, Moon Suk Kim

**Affiliations:** 1Department of Molecular Science and Technology, Ajou University, Suwon 16499, Korea; 2Department of Polymer Engineering, Pukyong National University, Busan 48547, Korea

**Keywords:** injectable hydrogel, depot, click-reaction, microsphere, retardation

## Abstract

Our purpose was to test whether a preparation of injectable formulations of dexamethasone (Dex)-loaded microspheres (Dex-Ms) mixed with click-crosslinked hyaluronic acid (Cx-HA) (or Pluronic (PH) for comparison) prolongs therapeutic levels of released Dex. Dex-Ms were prepared using a monoaxial-nozzle ultrasonic atomizer with an 85% yield of the Dex-Ms preparation, encapsulation efficiency of 80%, and average particle size of 57 μm. Cx-HA was prepared via a click reaction between transcyclooctene (TCO)-modified HA (TCO-HA) and tetrazine (TET)-modified HA (TET-HA). The injectable formulations (Dex-Ms/PH and Dex-Ms/Cx-HA) were fabricated as suspensions and became a Dex-Ms-loaded hydrogel drug depot after injection into the subcutaneous tissue of Sprague Dawley rats. Dex-Ms alone also formed a drug depot after injection. The Cx-HA hydrogel persisted in vivo for 28 days, but the PH hydrogel disappeared within six days, as evidenced by in vivo near-infrared fluorescence imaging. The in vitro and in vivo cumulative release of Dex by Dex-Ms/Cx-HA was much slower in the early days, followed by sustained release for 28 days, compared with Dex-Ms alone and Dex-Ms/PH. The reason was that the Cx-HA hydrogel acted as an external gel matrix for Dex-Ms, resulting in the retarded release of Dex from Dex-Ms. Therefore, we achieved significantly extended duration of a Dex release from an in vivo Dex-Ms-loaded hydrogel drug depot formed by Dex-Ms wrapped in an injectable click-crosslinked HA hydrogel in a minimally invasive manner. In conclusion, the Dex-Ms/Cx-HA drug depot described in this work showed excellent performance on extended in vivo delivery of Dex.

## 1. Introduction

Microspheres are capable of being loaded with several different types of drug compounds [1,2]. Drug-loaded microspheres can be subcutaneously injected to form a drug depot. The drug-loaded microsphere depots can sustain drug release over time. Thus, drug-loaded microsphere depots can significantly reduce dosage frequency and improve drug efficacy via a single injection [3,4,5,6].

To date, a number of techniques have been developed to prepare drug-loaded microspheres, including double emulsion, organic phase separation, and spray drying methods [7,8]. In previous studies, our group has manufactured drug-loaded microspheres using a monoaxial nozzle of an ultrasonic atomizer [9,10,11]. The advantages of this ultrasonic atomizer include a simple preparation process, high yield, and abundantly encapsulated drug. Additionally, several drugs have been loaded into the inner matrix of microspheres. Furthermore, drug-loaded microspheres can be easily prepared as an injectable drug-loaded microsphere formulation that can form a drug-loaded depot after subcutaneous injection, showing an in vivo drug release for at least four weeks in rats.

Nonetheless, the drug-loaded microsphere depot produces a relatively rapid drug release in the early days [9,10]. At least half of a drug is released in the first few days, although the drug release is sustained for four weeks. This initial burst may seem to be the result of a rapid release of molecularly dispersed drugs inside the external gel matrix layer throughout the microspheres and/or the drug released by cracked microspheres or through rapid perfusion of the releasing biological buffer into microsphere pores.

If the rapid drug release from microspheres in the early days can be suppressed, then the therapeutic drug level can be maintained for several such periods. Our strategy to achieve this aim was to wrap the microspheres with other matrix formulations, such as a hydrogel [12]. The drug initially shed by the microspheres may temporarily remain inside the external gel matrix, resulting in retardation of the drug release. Hence, the slowing of the initial release of the drug from microspheres can reduce the initial burst and maintain a suitable therapeutic level of release for several weeks. We therefore hypothesized that a hydrogel system, acting as an additional external gel matrix for the drug-loaded microspheres, can retard the rapid release of a drug from microspheres in the early days.

Among several hydrogel systems, injectable in situ-forming hydrogels can be prepared as a homogeneous suspension at ambient temperature [13,14,15,16,17,18]. Accordingly, drug-loaded microspheres can be mixed simply with an injectable in situ-forming hydrogel at ambient temperature. Then, a formulation of the in situ-forming hydrogel and drug-loaded microspheres can be introduced into the body in a minimally invasive manner.

Hyaluronic acid (HA) is used as a common ingredient in FDA-approved formulations for various medical applications [19,20,21,22]. An HA solution can be prepared as an injectable formulation and then injected to form a hydrogel depot at the injection site in a human tissue, but it quickly disappears under physiological conditions, even though HA cannot additionally form a hydrogel in vitro. The short residence time of the HA depot at an injection site can necessitate repeat doses due to the short service life of the drug depot.

Recently, we prepared a formulation of dexamethasone (Dex)-loaded microspheres (Dex-Ms) mixed with an HA solution [23]. Such formulations produced drug depots in a minimally invasive manner and yielded a sustained release pattern of the loaded Dex over a prolonged period. Nevertheless, HA gradually and completely disappeared under physiological conditions within six days.

Therefore, in this work, we studied injectable formulations with the aim of prolonging HA residence time to several weeks. It has been shown that a click reaction between transcyclooctene (TCO) and tetrazine (TET) rapidly forms a covalently crosslinked hydrogel without an external catalyst [24,25,26]. Thus, in this work, we designed TCO-modified HA (TCO-HA) and tetrazine-modified HA (TET-HA) and created crosslinked HA (Cx-HA) via the click reaction between TCO-HA and TET-HA to prolong HA residence time under physiological conditions.

Cx-HA may be a suitable candidate to prolong HA residence time. Thus, we hypothesized that in situ-forming Cx-HA hydrogel systems can retard the Dex release from Dex-Ms by acting as an additional external gel matrix for Dex-Ms. Consequently, in this study, we examined a pattern of the release of Dex from Dex-Ms-loaded Cx-HA (Dex-Ms/Cx-HA; Figure 1).

Meanwhile, for comparison, we chose commercially available Pluronic (PH), which consists of a central hydrophobic poly(propylene oxide) and two hydrophilic poly(ethylene oxide) segments. PH is the most widely used injectable in situ-forming hydrogel, although a PH hydrogel has short residence time under physiological conditions [27,28,29]. Thus, it is a suitable injectable in situ-forming hydrogel for comparison in this work.

The objectives of this study were to answer the following questions: (1) Can click-crosslinkable HA-based injectable formulations with Dex-Ms (Dex-Ms/Cx-HA) and injectable PH formulations with Dex-Ms (Dex-Ms/PH) be introduced into the body to produce a drug depot in a minimally invasive manner? (2) Does the Cx-HA or PH hydrogel have prolonged residence time under physiological conditions? (3) Can injectable formulations of Dex-Ms/Cx-HA retard the rapid release of Dex from Dex-Ms in the early days and maintain a therapeutic drug level during the desired period?

## 2. Materials and Methods

### 2.1. Materials

Poly(d,l-lactic-*co*-glycolic acid (PLGA: 50/50, Mw: 33,000 Da) was purchased from Birmingham Polymers, Inc. (Birmingham, AL, USA). Poly(vinyl alcohol) (PVA, 87–89% hydrolyzed, Mw: 85,000–124,000 Da), Dex, dimethyl sulfoxide (DMSO), maleic anhydride, fluorescein (FI), IR-783, 4-(4,6-dimethoxy-1,3,5-triazine-2-yl)-4-methyl-morpholinium chloride (DMTMM), sodium azide, propargyl amine, copper sulfate, ascorbic acid, and pyridine were bought from Sigma-Aldrich (St. Louis, MO, USA). Ethyl acetate, acetonitrile (ACN), toluene, and ether were used as received from Samchun (Gyeonggi, Korea). Dimethylformamide (DMF) were acquired from Junsei Chemical Co. (Chuo-ku, Tokyo), whereas methyltetrazine-PEG4-amine (TET) and trans-cyclooctene-amine (TCO) were from Click Chemistry Tools (Scottsdale, AZ, USA). Pluronic F-127 was used as received from BASF SE (BASF, Ludwigshafen, Germany). All other chemicals were of analytical grade and used without further purification. HA (1.0 MDa) was used as received from Humedix (Gyeonggi, Korea).

### 2.2. Preparation of Dex-Ms

A monoaxial one-nozzle atomizer (Sono-Tek Crop, Milton, NY, USA) was employed to prepare Dex-Ms. PLGA and Dex were dissolved at 3 wt % in ethyl acetate and at 1 wt % in DMSO, respectively. The PLGA and Dex solutions were mixed. Dex-Ms were produced by atomizing the mixed solutions of PLGA and Dex at flow rates of 4 mL/min and a vibration frequency of 3 W/60 kHz. The Dex-Ms were then collected in a 0.5% *w*/*v* PVA solution for 1 min. PVA can stabilize the microsphere formed from PLGA and Dex because hydroxyl groups in PVA can interact with the water phase, while the vinyl chain of PVA can interact with the PLGA and Dex in ethyl acetate. The distance between the atomizer head and the aqueous PVA solution was 1 cm, and the stirring speed of the PVA solution was 300 rpm. The resulting mixtures were gently stirred for 2 h at room temperature to allow for solidification of Dex-Ms and were then filtered and washed five times with deionized water (DW). The Dex-Ms were frozen at −74 °C, followed by freeze-drying over 4 days. The freeze-dried Dex-Ms were used for subsequent in vitro and in vivo experiments. Fluorescein-loaded microspheres (FI-Ms) were prepared from FI by the same procedures as described in the previous paragraph.

### 2.3. Encapsulation Efficiency of Dex-Ms

To determine the encapsulation efficiency of Dex, Dex-Ms (5 mg) were solubilized in 0.5 mL of ACN to dissolve the PLGA portion of the Dex-Ms and were sonicated for 60 min at 25 °C. Then, 0.5 mL of DW was added to the mixture and vortexed for 5 min at 25 °C. The amount of Dex in 1 mL of the mixed solution was determined using a high-performance liquid chromatography (HPLC) system (Agilent Technologies 1200 series, Waldbronn, Germany) with absorbance detection at 242 nm. Xterra RP 18 (3.9 × 150 mm, 5 μm, Waters, MA, USA) served as a chromatography column. The mobile phase consisted of ACN and DW (1:1, *v/v*) and was eluted at a flow rate of 1 mL/min. Three independent release experiments were performed for each Dex-Ms preparation. The encapsulation efficiency (E) was defined as follows: E = [(amount of encapsulated Dex)/(total amount of Dex added)] × 100.

### 2.4. Measurement of Dex-Ms Size

Each Dex-Ms sample (5 mg) was dispersed in DW (3 mL) and vortexed for 5 min at 25 °C. The particle size of each of the three independent Dex-Ms was determined by dynamic light scattering (ELSZ-1000; Otsuka Electronics, Osaka, Japan) at room temperature by following the manufacturer’s protocol. The size measurement was carried out in the instrument-associated software (ELS-Z, version 3.8, Otsuka, Japan).

### 2.5. Fabrication of a TET-HA and TCO-HA Hydrogel Formulation

The TET-HA was prepared by means of TET and HA. A 100 mg HA solution (0.25 mmol of COOH) was dispersed in 10 mL of phosphate-buffered saline (PBS; pH 7.4). DMTMM (20 mg, 0.07 mmol) and then TET (18.2 mg, 0.05 mmol) were individually added to the HA solution and stirred for 3 days at room temperature. For the preparation of transcyclooctene-amine-modified HA (TCO-HA), DMTMM (20 mg, 0.07 mmol) and then TCO (13.1 mg, 0.05 mmol) were individually added to the HA solution and stirred for 3 days at room temperature. The functionalized amount of TET and TCO was designed as 20% of total COOH on HA.

TET-HA and TCO-HA were dialyzed via a membrane with molecular weight cutoff (MWCO) 3.5–5 kDa for 3 days to remove unreacted TET or TCO. The obtained TET-HA and TCO-HA after dialysis were then lyophilized in a freeze dryer (FD 8508, Ilshinlab, Daejeon, Korea) for the next experiment. The yield of the obtained TET-HA and TCO-HA was defined as [(freeze-dried weight of TET-HA or TCO-HA after reaction)/(initial weight of (TET + HA) or (TCO + HA) before reaction)] × 100. To investigate the structure of the obtained TET-HA, TCO-HA, and Cx-HA, ^1^H nuclear magnetic resonance (NMR) spectra were recorded using a Mercury Plus 400 instrument (Varian, Palo Alto, CA, USA) with D_2_O. The resultant TET-HA and TCO-HA were individually used to make up a 2 wt % solution for subsequent in vitro and in vivo experiments.

### 2.6. Preparation of Hydrogel Formulation from Near-Infrared-Fluorescence-Labeled (NIR-Labeled) TET-HA and NIR-Labeled TCO-HA

IR-783 (250 mg, 0.33 mmol) was added into a 10 mL round flask, and 10 mL of DMF was then added. Sodium azide (30 mg, 0.5 mmol) was added to the IR-783 solution, which was then stirred at 65 °C for 24 h. Propargyl amine (42.6 mg, 0.66 mmol), copper sulfate (110 mg, 0.66 mmol), and ascorbic acid (240 mg, 1.32 mmol) were added into the IR-783-N3/DMF solution at room temperature, and the mixture was stirred for 1 day. The reaction solution was added into ether, and the resulting precipitates were filtered. The reacted solution was dried in vacuum to obtain brown IR783-NH2.

To synthesize NIR-labeled HA (NIR-HA), NIR-labeled TET-HA, or NIR-labeled TCO-HA, the COOH groups of 0.1 g of HA, TET-HA, or TCO-HA, the preparation of which is described in the previous subsection, were activated by the addition of DMTMM (16 mg, 0.057 mmol). After stirring at 25 °C for 1 h, NIR-783-NH2 (30 mg, 0.038 mmol) was added to the reaction solution, and the mixture was allowed to react for 24 h. The unreacted IR-783-NH2 was dialyzed via a membrane with MWCO 3.5–5 kDa for 3 days. In this work, HA solution was usually prepared in 2 wt % solution because its concentration above about 2 wt % showed high viscosity [30]. Thus, the prepared NIR-labeled HA, NIR-labeled TET-HA, or NIR-labeled TCO-HA were individually employed to make up a 2 wt % solution for subsequent in vivo imaging experiments.

### 2.7. Preparation of a NIR-Labeled PH (NIR-PH) Hydrogel Formulation

PH (0.5 g, 0.039 mmol) was added into a 100 mL round flask. Maleic anhydride (0.038 g, 0.38 mmol), 1 mL of pyridine, and 100 mL of toluene were added into the flask and then stirred for 8 h at room temperature. The reaction mixture was poured into ether to precipitate PH–COOH. PH–COOH (0.1 g, 1.2 mmol) was solubilized in 30 mL of DW and activated by the addition of DMTMM (16 mg, 0.057 mmol). After stirring at 25 °C for 1 h, the IR-783-NH2 (30 mg, 0.038 mmol), the preparation of which is described in the previous subsection, was added into the solution, which was then stirred for 1 day at room temperature. The unreacted IR-783-NH2 was removed by dialysis via a membrane with MWCO 3.5–5 kDa for 3 days. The quantitative functionalization of the COOH group was determined by protons of vinyl group for PH–COOH in ^1^H NMR [31]. The prepared NIR-labeled PH was used to make up a 20 wt % solution for a subsequent in vivo imaging experiment because PH has been widely used in 20 wt % as injectable in situ-forming hydrogel [32].

### 2.8. Preparation of Dex-Ms, Dex-Ms-Loaded PH, and Dex-Ms-Loaded Cx-HA

The prepared Dex-Ms (2.5 mg in Dex terms) were separately added to the 0.5 mL of TET-HA (20 mg/mL) and 0.5 mL of TCO-HA (20 mg/mL) solutions. Dex-Ms-loaded TET-HA (0.5 mL) and Dex-Ms-loaded TCO-HA (0.5 mL) solutions were mixed in a vial to prepare Dex-Ms-loaded Cx-HA (5 mg in Dex terms), which was stored at 4 °C for 48 h. PH was dissolved in PBS (pH 7.4) at 20 wt % and incubated at 4 °C. Dex-Ms (5 mg in Dex terms) were added to the PH solution to prepare Dex-Ms-loaded PH and were vortexed for 5 min and then stored at 4 °C for 48 h. We will refer to Dex-Ms-loaded Cx-HA and Dex-Ms-loaded PH as Dex-Ms/Cx-HA and Dex-Ms/PH, respectively, for convenience. Large amounts of Dex-Ms/Cx-HA and Dex-Ms/PH were prepared for rheological measurements and an in vitro Dex release in a subsequent experiment.

### 2.9. Rheological Analysis of the Dex-Ms/Cx-HA and Dex-Ms/PH

Rheological properties of Dex-Ms/Cx-HA and Dex-Ms/PH, the preparation of which is described in the previous subsections, were measured using a modular compact rheometer (MCR 102, Anton Paar, Austria) with a Peltier temperature-controlled bottom platen. The parallel-plate diameter was 25 mm. All the measurements were conducted with a gap length of 0.3 mm at a frequency of 1 Hz and 1% strain at 25 and 37 °C. The storage modulus (G’) and loss modulus (G”), viscosity (Pa·s), and the phase angle (tan δ) were calculated by the instrument’s software (Rheoplus/32, version V3.21, Anton Paar, Graz, Austria). Each set of samples was repeated three times.

### 2.10. In Vitro Persistence of Injectable Formulation by Rheological Analysis

Dex-Ms alone, Dex-Ms/PH, or Dex-Ms/Cx-HA, the preparation of which is described in the previous subsections, were shaken at 100 rpm in an incubator at 37 °C. At the specified time points of sample collection, rheological properties of all formulations were measured using a modular compact rheometer.

### 2.11. In Vitro Persistence of Injectable Formulation by Weight Change

To measure the weight change of all formulations, the prepared PH, Dex-Ms/PH, Cx-HA, or Dex-Ms/Cx-HA (as described in Section 2.8) were individually added into a 15 mL vial. Then, 5 mL of PBS was added, and the mixtures were incubated at 37 °C. At predetermined time points, PBS was carefully decanted from the vials. After that, each PH, Dex-Ms/PH, Cx-HA, or Dex-Ms/Cx-HA in vials was lyophilized until the residue reached a constant weight in a freeze dryer. The vial was weighed to determine the dried weight of PH, Dex-Ms/PH, Cx-HA, or Dex-Ms/Cx-HA at predetermined time points. The ratio of remaining weight was defined as follows: Remaining weight (%) = [(initial weight − dried weight at predetermined time points)/(initial weight)] × 100.

### 2.12. In Vitro Dex Release

Dex-Ms alone (5 mg by Dex), Dex-Ms/PH (5 mg/mL by Dex), and Dex-Ms/Cx-HA (5 mg/mL by Dex), the preparation of which is described in the previous subsections (Section 2.2, Section 2.7 and Section 2.8), were transferred to fresh 5 mL vials, resuspended in 4 mL of PBS (pH 7.4), and shaken at 100 rpm in an incubator at 37 °C. At the specified time points of sample collection, 0.5 mL of the solution was removed from each vial and replaced with 0.5 mL of fresh PBS (maintained at 37 °C). The amount of a Dex release was determined by HPLC as described earlier. Three independent release experiments were conducted for each sample. The cumulative amount of Dex released in vitro was calculated by comparison with standard calibration curves constructed with known concentrations of Dex.

### 2.13. In Vivo Dex Release

All Sprague Dawley rats (280–300 g, 6 weeks, male) were performed in accordance with the guidelines of the Institutional Animal Experiment Committee at Ajou University School of Medicine (approval No. 2016-0048, 12 October 2016). All animals used in this work were treated in accordance with the approved guidelines for the Care and Use of Animals for Experimental and Scientific Purposes.

Dex-Ms alone (5 mg/mL Dex concentration) formulation was prepared at concentration of 20% (*w*/*v*) in mixture of 5% D-mannitol, 2% carboxymethylcellulose, and 0.1% Tween 80 and then loaded into a 1 mL syringe. A Dex-Ms /PH (5 mg/mL Dex concentration) formulation was prepared from 0.3 mL of the PH solution and then loaded into a 1 mL syringe. For preparation of the Dex-Ms/Cx-HA formulation (5 mg/mL Dex concentration), Dex-Ms (2.5 mg/mL Dex concentration) was separately added into 0.15 mL of the TET-HA solution and 0.15 mL of the TCO-HA solution. The Dex-Ms-loaded TET-HA solution and Dex-Ms-loaded TCO-HA solution were separately loaded into each barrel of a dual-barrel syringe.

Dex-Ms alone, Dex-Ms/PH, Dex-Ms-loaded TET-HA solution, and Dex-Ms-loaded TCO-HA solution were sterilized with UV irradiation overnight. The rats were randomly distributed into three experimental groups: Dex-Ms alone (*n* = 21), Dex-Ms/PH (*n* = 21), and Dex-Ms/Cx-HA (*n* = 21). Each rat was anesthetized using Zoletil and Rompun (1:1 ratio, 1.5 mL·kg^−1^). Next, 300 μL of Dex-Ms alone or of the Dex-Ms/PH formulation loaded into a 1 mL syringe was injected individually and subcutaneously under the dorsal skin of each experimental rat using a 21-gauge (21 G) needle. The Dex-Ms-loaded TET-HA solution and Dex-Ms-loaded TCO-HA formulation separately loaded into a dual-barrel syringe (total 300 μL, 150 μL each) were injected individually and subcutaneously under the dorsal skin of each experimental rat using a 21 G needle.

The rats were euthanized 1, 3, 5, 10, 14, 21, or 28 days after the injection (experimental time points designed to observe Dex release as 1, 3, 5, and 10 days for initial period and 14, 21, and 28 days for long term). All formulation implants were removed individually from the subcutaneous dorsum. All the removed formulation implants were freeze-dried for 5 days and then homogenized by means of a T10 basic ULTRA-TURRAX Homogenizer (IKA, Werke GMBH, Staufen, Germany) at 25,000–30,000 rpm for 10 min. After that, 0.5 mL of ACN was added into each homogenized sample, and the mixture was sonicated for 1 day. Next, 1 mL of DW was added into the mixture and centrifuged at 2000 rpm for 10 min. The amount of Dex in the DW-soluble portion was determined by HPLC as described in a previous subsection. The amount of released Dex was defined as follows: Released Dex amount = [injected Dex amount − determined Dex amount at predetermined time points]. The released Dex amount was calculated by accumulation of Dex determined at predetermined time points. Three independent experiments for in vivo Dex release were conducted for each formulation implant.

### 2.14. In Vivo Persistence via Fluorescence Imaging

To monitor a real-time in vivo release in live animals, 150 μL of FI-Ms alone, NIR-PH alone, NIR-HA alone, or FI-M-loaded NIR-labeled PH (FI-M-NIR-PH) formulation loaded into a 1 mL syringe was individually injected subcutaneously under the dorsal skin of nude mice using a 21 G needle. The FI-M-loaded NIR-labeled TET-HA solution and FI-M-loaded NIR-labeled TCO-HA solution, which were separately loaded into each barrel of a dual-barrel syringe (total 150 μL, 75 μL each) were injected subcutaneously under the dorsal skin of nude mice using a 21 G needle.

For defined periods, side view images of the mice were captured at a wavelength of fluorescence excitation filtered with a 460–730 nm band-pass filter; for emission, light was filtered with a 500–525 nm and 750–825 nm band-pass filters. At each time point, fluorescent images were acquired on an imaging instrument (FOBI, NeoScience, Suwon, Korea). NIR fluorescence photos were taken with exposure time of 500 ms and gain of one using a dichroic cube filter (MgF2, fused silica filter).

### 2.15. SEM Analysis

In vivo-injected hydrogel implants Dex-Ms, Dex-Ms/PH, and Dex-Ms/Cx-HA were removed individually from the subcutaneous dorsum of each rat at 1 week. The morphology was examined by SEM (JSM-6700F, JEOL, Tokyo, Japan). The removed in vivo formulation implants were rapidly immersed in a liquid nitrogen bath to prevent structural changes and then lyophilized in a freeze dryer at −75 °C for 5 days. Each dried formulation implant was coated with a thin layer of gold in an argon atmosphere using a plasma-sputtering apparatus (Ted Pella, Cressington 108 Auto, Redding, CA, USA) and then examined by SEM.

### 2.16. Histological Analysis

The rats implanted with Dex-Ms alone, Dex-Ms/PH, or Dex-Ms/Cx-HA were euthanized at 1 and 4 weeks. The removed implants were immediately fixed with 10% formalin. The paraffin wax-embedded specimens were sectioned at 4 μm thickness along the longitudinal axis of the implant on an HM 340E electronic rotary microtome (Thermo Fisher Scientific, Waltham, MA, USA).

For hematoxylin and eosin (H&E) staining, the implants were incubated at 60 °C for 1 day to remove paraffin. The deparaffinized sections were washed with xylene and then hydrated using 100%, 95%, 70%, and 60% ethyl alcohol in this sequence. Then, the slides were stained with a hematoxylin solution (Sigma, St. Louis, MO, USA) and eosin solution (Sigma, St. Louis, MO, USA) for 3 min, respectively. Thereafter, the stained slides were fixed and mounted with a mounting medium (Muto Pure Chemicals, Tokyo, Japan).

For 4′,6-diamino-2-phenylindole dihydrochloride (DAPI; Sigma-Aldrich, St. Louis, MO, USA) and macrophage (ED1) staining, the slides were incubated for 10 min at 120–130 °C in citrate buffer (Sigma, St. Louis, MO, USA) and then for 10 min in PBS. The slides were washed two times for 5 min with PBS containing 0.05% Tween 20 (PBST, Sigma, St. Louis, MO, USA). The slides were next blocked with a 5% solution of bovine serum albumin (BSA; Bovogen, Keilor East, Australia) containing 5% horse serum (HS; Gibco, Invitrogen, Carlsbad, CA, USA) in PBS for 90 min at 37 °C. The slides were exposed to a mouse anti-rat CD68 antibody (ED1; Serotec, Oxford, UK) in an antibody diluent (DAKO, Glostrup, Denmark) (1:1000) with incubation for 15 h at 4 °C. The slides were rinsed with PBS for 5 min and PBST for 10 min and then incubated with the secondary antibody (goat anti-mouse IgG antibody conjugated with Alexa Fluor 594; Invitrogen) (1:200) for 3 h in the dark at room temperature. The slides were washed with PBST and mounted by means of the Pro-Long Gold Antifade Reagent with DAPI (Life Technologies, Grand Island, NY, USA). Immunofluorescent images were captured with an Axio Imager A1 (Carl Zeiss Microimaging GmbH, Göttingen, Germany) equipped with Axiovision Rel. 4.8 software (Carl Zeiss).

### 2.17. Statistical Analysis

All data on the encapsulation of Dex-Ms, particle size of the Dex-Ms, rheological measurements, in vitro and in vivo release, and ED1 assay were obtained from three independent experiments. All the data are presented as means ± SD. The data from each experiment were examined by one-way ANOVA with Bonferroni’s multiple comparison test. Statistical analyses were performed in the SPSS 12.0 software (IBM Corporation, Armonk, NY, USA).

## 3. Results and Discussion

### 3.1. Preparation of Dex-Ms

In this work, an ultrasonic atomizer was employed to prepare Dex-Ms (Figure 2). A mixture of PLGA and Dex was introduced into the ultrasonic atomizer. The atomization process resulted in the formation of microdroplets in PVA solution during passage through the nozzle. Then, Dex-Ms were obtained by the solvent evaporation/extraction method via spraying of microdroplets over water containing PVA. Images of the prepared Dex-Ms revealed a spherical shape and smooth surface. The cross-sectioned SEM image showed the inner surface was filled, not hollow (Figure 2e’).

Dex-Ms were obtained with a yield of 85%, an encapsulation efficiency of 80%, and a particle size of 57 ± 27 μm for subsequent in vitro and in vivo release experiments involving subcutaneous injection into rats. FI-Ms, which were prepared to examine real-time live release images, were greenish due to the color of FI. FI-Ms had a size, surface, and shape similar to those of Dex-Ms.

### 3.2. Preparation and Characterization of Injectable Dex-Ms-Loaded Hydrogel Formulations

To prepare the Cx-HA hydrogel, the carboxylic groups of HA were first activated and then separately reacted with TET and TCO to obtain TET-HA and TCO-HA. The reactions showed a >90% yield of TET-HA and TCO-HA. Elemental analysis of the amine groups in TET-HA and TCO-HA showed that there was quantitative introduction of TET and TCO [33]. ^1^H NMR spectra showed the characteristic peaks of TET and TCO modified into HA (Figure 2f). In ^1^H NMR spectra, new characteristic peaks (5 and 6) of Cx-HA hydrogel were observed, and TET and TCO disappeared after the click reaction between TET-HA and TCO-HA.

Next, Dex-Ms alone were easily prepared as an injectable formulation (see the experimental part) (Figure 3a). The prepared TET-HA and TCO-HA were dissolved in PBS at a concentration of 2% (*w*/*v*). The TET-HA and TCO-HA solutions were immediately click-crosslinked via mixing of their equal amounts, resulting in the formation of the Cx-HA hydrogel. The TCO-HA solution was translucent and colorless, but the TET-HA solution was slightly reddish due to the original color of TET (Figure 3b).

A PH solution was prepared in deionized water at 20 wt %. The PH immediately became a hydrogel depot at 37 °C but completely turned into a liquid after two days (Figure 3c).

A Dex-Ms-loaded PH formulation (Dex-Ms/PH) was easily prepared by mixing of Dex-Ms and PH solutions (Figure 3d). Dex-Ms/PH was a liquid solution at 25 °C and immediately became a hydrogel depot at 37 °C. To examine in vitro persistence of the injectable Dex-Ms/PH formulation, Dex-Ms/PH was incubated at 37 °C. Dex-Ms/PH remained a stable hydrogel depot for approximately 20 h. Nonetheless, after incubation at 37 °C for two days, the Dex-Ms/PH depot completely turned into a liquid.

Dex-Ms were individually mixed with the TET-HA solution and TCO-HA solution (Figure 3e). A Dex-Ms/Cx-HA depot immediately formed after mixing of equal amounts of the Dex-Ms-loaded TET-HA solution and the Dex-Ms-loaded TCO-HA solution because the click-crosslinking reaction between TET and TCO proceeded quickly. The formed Dex-Ms/Cx-HA depot then persisted in an almost stable depot form for at least 14 days, indicating that the Dex-Ms/Cx-M depot preserved its structural gelatinous integrity. Thereafter, the size of Dex-Ms/Cx-HA depot gradually decreased by day 28. These results indicated that Cx-HA could act as a depot for Dex-Ms for 28 days.

The weight changes of hydrogel depot at 37 °C were determined to examine in vitro persistence of injectable formulations (Figure 3f). PH, Dex-Ms/PH, Cx-HA, or Dex-Ms/Cx-HA was incubated at 37 °C. The weight changes of PH and Dex-Ms/PH hydrogel depots gradually decreased, reaching almost zero at 50 h, indicating that PH and Dex-Ms/PH hydrogel depots got completely solubilized in PBS within 50 h. In contrast, Cx-HA and Dex-Ms/Cx-HA hydrogel depots showed weight changes of only 5% for 168 h. This result indicated that Cx-HA hydrogel without and with Dex-Ms acted as a hydrogel depot during a defined experimental period.

Next, rheological characterization of each formulation was performed (Figure 4). The storage and loss moduli of TET-HA and TCO-HA were 0.3–0.5 Pa. The storage and loss moduli of Cx-HA were 192 Pa and 110 Pa, respectively (Figure 4a). TET-HA and TCO-HA solutions had viscosity of 0.1 Pa·s, but Cx-HA formed by the mixing of TET-HA and TCO-HA solutions had sharply higher viscosity of 131 Pa·s (Figure 4b), confirming the click-crosslinking reaction between TET and TCO. Cx-HA manifested higher storage and loss moduli as well as viscosity than TET-HA or TCO-HA, indicating high hydrogel-like stiffness of Cx-HA compared with TET-HA or TCO-HA before click-crosslinking. Additionally, the formed Cx-HA had almost the same viscosity between 25 and 37 °C. Meanwhile, the PH solution had 139 Pa·s viscosity at 25 °C and much higher viscosity of 910 Pa·s at 37 °C, indicating a phase transition at body temperature (Figure 4c).

PH, Dex-Ms/PH, Cx-HA, and Dex-Ms/Cx-HA showed little difference in storage and loss moduli (Figure 4d) or viscosity (Figure 4e) without and with Dex-Ms at 37 °C. This finding indicated that Dex-Ms did not affect the viscosity of the hydrogel.

Next, the viscosity of PH, Dex-Ms/PH, Cx-HA, or Dex-Ms/Cx-HA were compared according to incubation for 28 days at 37 °C to examine in vitro persistence of injectable formulations (Figure 4f). Shortly after incubation of Dex-Ms/PH and Dex-Ms/Cx-HA, the viscosity of the Dex-Ms/PH hydrogel became higher than that of Dex-Ms/Cx-HA. Nonetheless, the viscosity of the Dex-Ms/PH hydrogel was not measured at two days owing to the dissipation of PH. Meanwhile, the viscosity of Dex-Ms/Cx-HA manifested almost constant values up to eight days and gradually decreased by day 28. These results suggested that Cx-HA in Dex-Ms/Cx-HA could act as a stable hydrogel during a defined experimental period.

### 3.3. An In Vitro Dex Release from Injectable Formulations

The in vitro patterns of Dex release from Dex-Ms alone, Dex-Ms/PH, or Dex-Ms/Cx-HA were examined at 37 °C for 28 days (Figure 5). The cumulative amount of Dex released by Dex-Ms alone was 27% on day 5, owing to an initial burst, and then a 40% release of Dex by day 8. After eight days, Dex-Ms alone manifested a linear release pattern up to 77% for 28 days.

For Dex-Ms/PH, the Dex release was 4.5% on day 3 and 6.8% on day 4, indicating the retardation of Dex release in the early period. Thereafter, the cumulative amount of Dex was 12.8% on day 5, 19.8% on day 6, and then 26% on day 8. This release retardation pattern can probably be explained as follows: Dex remained in some lipophilic portions of PH and interacted with lipophilic portions of PH in the early period, and increased Dex release was therefore observed due to the dissipation of PH in the release medium at 37 °C. The cumulative amount of Dex showed a linear release pattern up to 72% for 28 days.

For Dex-Ms/Cx-HA, the Dex release was 6.6% on day 3 and 9.3% on day 4, indicating the slowing of Dex liberation in the initial period. The amounts released by Dex-Ms/Cx-HA were slightly higher than those of Dex-Ms/PH for the same period until day 4. This was due to the slightly higher viscosity of the PH hydrogel compared with the HA hydrogel, as described in a previous subsection, and/or innate hydrophilic property of Cx-HA. After five days, the cumulative amount of Dex shed by Dex-Ms/Cx-HA manifested a linear release pattern up to 34% for 28 days, indicating remarkable retardation of Dex liberation. This effect was due to slowing of the Dex release by the Cx-HA hydrogel acting as an external gel matrix of Dex-Ms.

The remaining amounts of Dex in Dex-Ms alone, Dex-Ms/PH, and Dex-Ms/Cx-HA were found to be 18%, 25%, and 57% at 28 days, respectively. The sum of the shed and remaining Dex amounts was almost the same as the Dex amount loaded inside Dex-Ms alone, Dex-Ms/PH, and Dex-Ms/Cx-HA. This result indicated that microspheres acted as a depot of Dex at first, and the hydrogel then served as a depot, acting as an external gel matrix of the microspheres.

### 3.4. In Vivo Depot Formulation and Persistence of Injectable Materials

First, to confirm the formation of an in vivo depot, Dex-Ms and Dex-Ms/PH formulations were each drawn into a single syringe, whereas Dex-Ms-loaded TET-HA and Dex-Ms-loaded TCO-HA formulations were separately drawn into each compartment of a dual-barrel syringe (Figure 6a). Dex-Ms and Dex-Ms/PH formulations formed depots almost immediately after injection via a 21 G needle. Dex-Ms/TET-HA and Dex-Ms/TCO-HA inside each compartment of a dual-barrel syringe were also injected into rats using a 21 G needle to allow for formation of an in vivo click-crosslinked Dex-Ms/Cx-HA depot within a few seconds (Figure 6b).

To investigate in vivo formation of a drug depot, implants in tissues were excised from rats after one week and then examined. In optical images (Figure 6c), the removed implants showed a gelatinous hydrogel form with blood vessels around the hydrogel implants. The sizes of Dex-Ms or Dex-Ms/PH were smaller than the size of Dex-Ms/Cx-HA. The decreased size of Dex-Ms/PH was probably due to the in vivo dissipation of the PH hydrogel, as described in a previous subsection. In SEM images (Figure 6d), the Dex-Ms appeared to have a spherical shape, implying the formation of Dex-Ms drug depot.

Next, we examined the in vivo persistence of the injectable materials by real-time imaging. The same amount of the NIR fluorescent label was chemically introduced into PH alone, HA alone, and into in vivo click-crosslinked Cx-HA from TET-HA and NIR-TCO-HA.

FI-Ms alone, NIR-PH alone, NIR-HA alone, and FI-M/NIR-PH formulations were each drawn into a single syringe (Figure 7a). Additionally, FI-M-loaded NIR-TET-HA and FI-M-loaded NIR-TCO-HA formulations were separately drawn into each compartment of a dual-barrel syringe (Figure 7b). All the formulations were injected separately into subcutaneous tissue of a mouse and formed a depot almost immediately after injection.

After the injection of FI-Ms alone (Figure 7c), the fluorescent images showed an almost unchanged signal between 6 h and two days. The intensity of fluorescent images gradually decreased up to two weeks. Almost no fluorescent signals were detected at 21 days. This result indicated that FI-Ms alone yielded a sustained fluorescent image for at least two weeks, although Dex and FI have different release patterns due to the different structures. PLGA generally degraded through in vivo hydrolytic chain scission of PLGA [9,10]. The microsphere in this work gradually degraded for four weeks (data not shown).

For the NIR-PH-only formulation (Figure 7d), NIR signals of NIR-PH (red) were seen in images at 6 h after injection. The NIR signals of NIR-PH decreased on day 1 and then almost disappeared after three days, indicating rapid clearance of NIR-PH from subcutaneous tissue.

In the case of the NIR-HA-only formulation (Figure 7e), NIR signals of NIR-HA (red) were clearly visible in images at 6 h after injection. The NIR signals of NIR-HA decreased on day 1 and then almost disappeared after three days. This result indicated that NIR-PH and NIR-HA disappeared from subcutaneous tissue at almost the same rate.

In the case of FI-Ms/NIR-PH (Figure 7f), red images of NIR-PH (red) were well visible on day 1, and some degree of a yellow image (merged red and green) was seen on day 1. The yellow signals gradually decreased up to two days owing to the dissipation of the PH hydrogel in vivo. Thereafter, green signals attributable to FI-Ms appeared and then gradually decreased up to 21 days (as depicted in Figure 7c). The FI-Ms/NIR-PH yielded green images similar to those of FI-Ms alone, indicating that NIR-PH slowed down the FI release from FI-Ms only in the early period of 2–3 days.

Meanwhile, injection of FI-Ms-loaded NIR-TET-HA and FI-Ms-loaded NIR-TCO-HA from a dual-barrel syringe afforded red and weak yellow signals (merged red and green) at 6 h after injection (Figure 7g), indicating formation of an in vivo FI-Ms-Cx-HA depot. The yellow signals increased at 6 h and one day and thereafter gradually decreased from the second day to two weeks. Nevertheless, slight yellow signals were observed even at three weeks.

NIR (red) signals assignable to NIR-Cx-HA were also observed at 6 h after injection (Figure 7h). NIR signals of NIR-Cx-HA persisted in vivo for four weeks, although NIR signals gradually decreased due to the degradation of Cx-HA. This result indicated that FI-Ms-loaded NIR-TET-HA and FI-Ms-loaded NIR-TCO-HA formulations formed a depot via click-crosslinking and that the formed FI-Ms-loaded NIR-Cx-HA depot sustained a stable in vivo release for at least four weeks.

The fluorescent signals (green) from the FI-Ms-only depot could not be detected at 21 days (Figure 7c); however, green fluorescent signals from the FI-Ms-loaded NIR-Cx-HA depot were seen at 6 h and gradually decreased up to four weeks (Figure 7i), indicating that the FI intensity was maintained for an extended period in the FI-Ms-loaded NIR-Cx-HA depot. This result strongly suggested that the reduction of FI green fluorescent signals was slowed by the NIR-Cx-HA hydrogel as an additional external gel matrix for the FI-Ms. On the basis of these results, we confirmed that NIR-Cx-HA successfully acted as an outer persistent drug depot for the FI-Ms (Dex-Ms) over the defined experimental period.

### 3.5. In Vivo Dex Release from Injectable Formulations

To examine the in vivo Dex release, Dex-Ms alone, Dex-Ms/PH, or Dex-Ms/Cx-HA was injected into the subcutaneous tissue of Sprague Dawley rats. The released amount of Dex was determined by means of the remaining Dex amount in the excised Dex-Ms-only, Dex-Ms/PH, or Dex-Ms/Cx-HA depots at each experimental time point. Figure 8 shows a plot of the liberated Dex amount versus release time.

The released amount of Dex from Dex-Ms alone was significantly higher (18% of the total) in comparison with other formulations on day 1, then the release of 44% was detected up to seven days. Thereafter, the release of Dex reached 83% at 28 days. These data indicated that half of Dex was released in the first seven days compared with the total liberated amount after 28 days.

In the case of Dex-Ms/PH, the released amount of Dex was much lower (3.4%) on day 1 and 13.6% on day 3, probably due to the interaction of Dex with lipophilic portions of PH. The linear release pattern of Dex was observed for up to seven days (32% of the total). This result was due to the dissipation of the PH hydrogel within three days in vivo. The release pattern was consistent with the in vitro release and NIR imaging examined in the previous subsection. Thereafter, the released amount of Dex reached 78% at 28 days, and the release pattern was similar to that of Dex-Ms alone.

For Dex-Ms/Cx-HA, the Dex release was 4.7% on day 1 and 14.4% on day 3. The in vivo released amounts from Dex-Ms/Cx-HA were slightly higher than those of Dex-Ms/PH for the same period until three days. This finding is due to the slightly higher viscosity of the PH hydrogel compared with the Cx-HA hydrogel and/or innate hydrophilic property of Cx-HA as described in a subsection above. After five days, the in vivo cumulative amount of Dex released by Dex-Ms/Cx-HA showed a decrease to 42% (relative to Dex-Ms/PH) at 28 days, indicating remarkable retardation of the Dex release. This result was due to the slowing of the Dex release by the Cx-HA hydrogel acting as an external gel matrix of Dex-Ms. The in vivo release pattern was in good agreement with the in vitro release pattern examined in a subsection above.

Several investigators, including our group, have reported that Dex-Ms could achieve Dex release over a period of one month. However, most results were examined for in vitro Dex release, and there are little or no in vivo Dex release results [12,22,34]. Additionally, in order to obtain the desired release profile, the decrease in the initial Dex burst release was examined by several parameters [35]. To the best of our knowledge, injectable formulations of Dex-Ms/Cx-HA to sustain Dex release by external gel matrix over an extended period have received little attention in the literature. In this work, the in vivo Dex release sustained a stable release for at least four weeks and was in good agreement with the in vivo NIR fluorescence examined in a subsection above.

Collectively, these results support the main hypothesis of this study, i.e., that injectable formulations of Dex-Ms/Cx-HA retard the rapid release of Dex from Dex-Ms in the early days and maintain a therapeutic drug level during the desired period.

### 3.6. Host Tissue Response to the Injectable Materials

The depot implants resulting from the injection of Dex-Ms alone, Dex-Ms/PH, or Dex-Ms/Cx-HA was stained with an anti-ED1 antibody to compare inflammatory responses (Figure 9a). Tissues stained with the anti-ED1 antibody (red) and with DAPI (blue) revealed accumulation of inflammatory cells and the extent of host cell infiltration within and near the injected depot, respectively.

ED1-positive cells were found at the surface and in tissues surrounding the injected depot, indicating macrophage accumulation. ED1-positive cells were counted in the stained tissue area (Figure 9b). The percentages of ED1-positive cells in depots of Dex-Ms alone, Dex-Ms/PH, or Dex-Ms/Cx-HA was 33%, 27%, and 25%, respectively, after one week. After four weeks, the corresponding percentages of ED1-positive cells decreased to 9%, 4.6%, and 4%. The inflammation caused by the depot of Dex-Ms alone was slightly higher in comparison with Dex-Ms/PH and Dex-Ms/Cx-HA. This finding was probably due to the biocompatibility of the hydrogels acting as an external gel matrix of Dex-Ms.

## 4. Conclusions

In summary, we confirmed that a depot of Dex-Ms wrapped with an injectable click-crosslinkable HA hydrogel successfully formed at the injection site in rats in a minimally invasive manner. Cx-HA in the Dex-Ms/Cx-HA formulation acted as a persistent hydrogel with residence time longer than that of Dex-Ms/PH. Additionally, the Dex-Ms/Cx-HA formulation retarded the rapid release of Dex from Dex-Ms in the early days and thus maintained a therapeutic drug level for over four weeks. Therefore, the injectable formulation based on microspheres wrapped in a hydrogel prepared in this work may help to maintain a therapeutic drug level for an extended period.

## Figures and Tables

**Figure 1 pharmaceutics-11-00438-f001:**
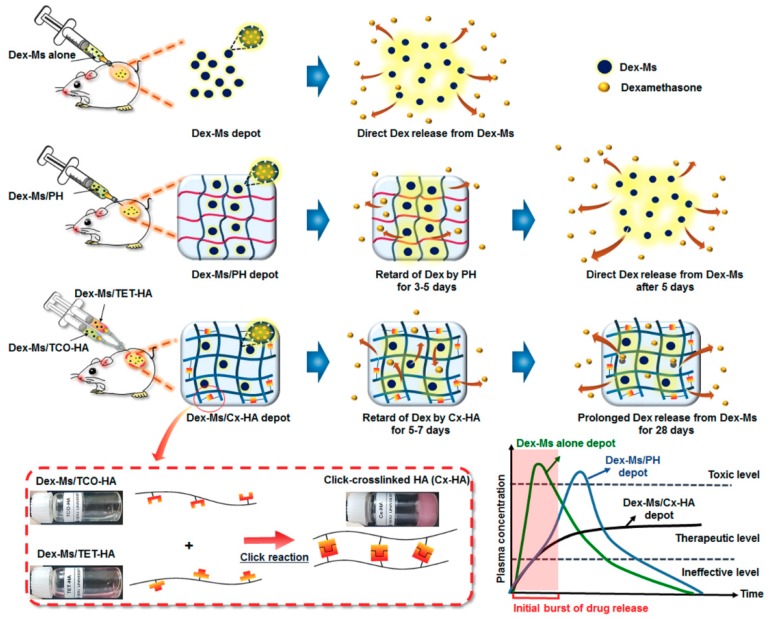
Schematic representation of a prolonged drug release from dexamethasone (Dex)-loaded microsphere (Dex-Ms) depot using injectable Pluronic (PH) and click-crosslinked hyaluronic acid (Cx-HA) hydrogels and injectable Cx-HA between tetrazine (TET)-modified HA (TET-HA) and transcyclooctene (TCO)-modified HA (TCO-HA). (Images were drawn by J.H.N. and S.H.P. in the Adobe Photoshop 7.0 software.).

**Figure 2 pharmaceutics-11-00438-f002:**
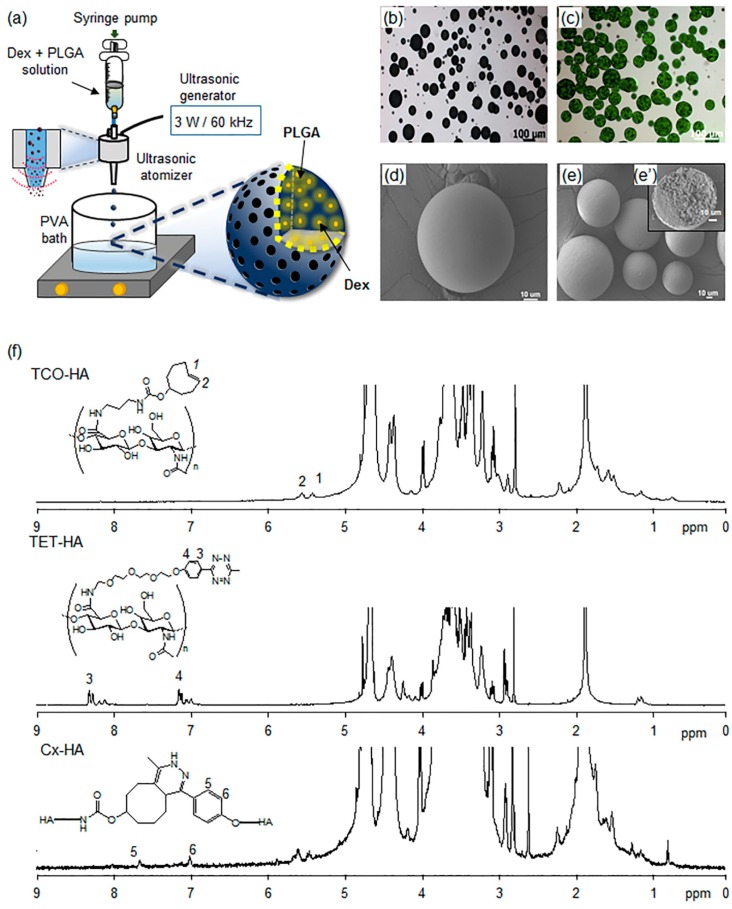
(**a**) Schematic representation of the encapsulation method using a monoaxial one-nozzle ultrasonic atomizer; (**b**) optical microscopy of Dex-Ms; (**c**) fluorescence microscopy of fluorescein (FI)-loaded microspheres (FI-Ms); (**d**,**e**,**e’**) scanning electron microscopy of (**d**,**e**) Dex-Ms and (**e’**) cross-sectioned Dex-Ms [magnification is ×1000 (**d**) and ×400 (**e**,**e’**)]; and (**f**) ^1^H NMR spectra of TET-HA, TCO-HA, and Cx-HA. (Image (**a**) was drawn by J.Y.H. and J.H.N. in the Adobe Photoshop 7.0 software).

**Figure 3 pharmaceutics-11-00438-f003:**
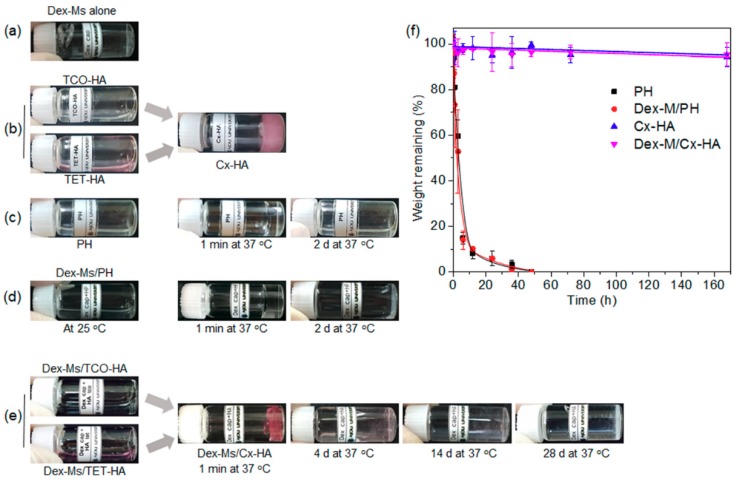
Images for the injectable formulations. (**a**) Dex-Ms alone, (**b**) TET-HA, TCO-HA, and Cx-HA, (**c**) PH at 25 and 37 °C after 1 min and two days, (**d**) Dex-Ms/PH at 25 and 37 °C after 1 min and two days, (**e**) Dex-Ms/TET-HA, Dex-Ms/TCO-HA, and Dex-Ms/Cx-HA at 37 °C for 28 days, and (**f**) the remaining weight of PH, Dex-Ms/PH, Cx-HA, and Dex-Ms/Cx-HA at 37 °C versus time. (All data were performed three times and are presented as the mean and standard deviation).

**Figure 4 pharmaceutics-11-00438-f004:**
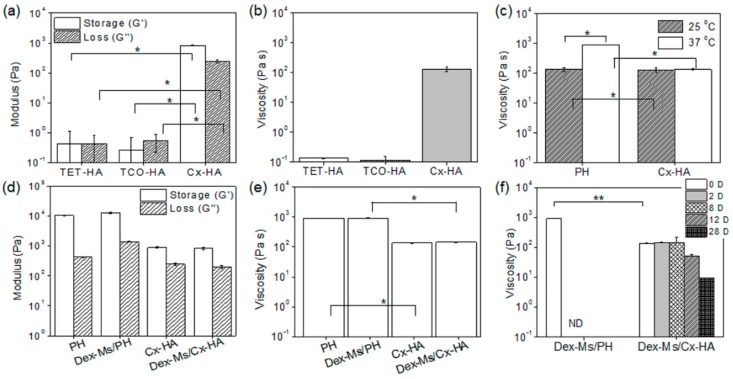
(**a**) Storage and loss modulus and (**b**) viscosity of TET-HA, TCO-HA, and Cx-HA; (**c**) PH and Cx-HA at 25 and 37 °C; (**d**) storage and loss modulus and (**e**) viscosity of PH, Dex-Ms/PH, Cx-HA, and Dex-Ms/Cx-HA at 37 °C; and (**f**) Dex-Ms/PH and Dex-Ms/Cx-HA at 1, 2, 8, 12, and 28 days at 37 °C (* *p* < 0.001, ** *p* < 0.05). (All data were performed three times and are presented as the mean and standard deviation.).

**Figure 5 pharmaceutics-11-00438-f005:**
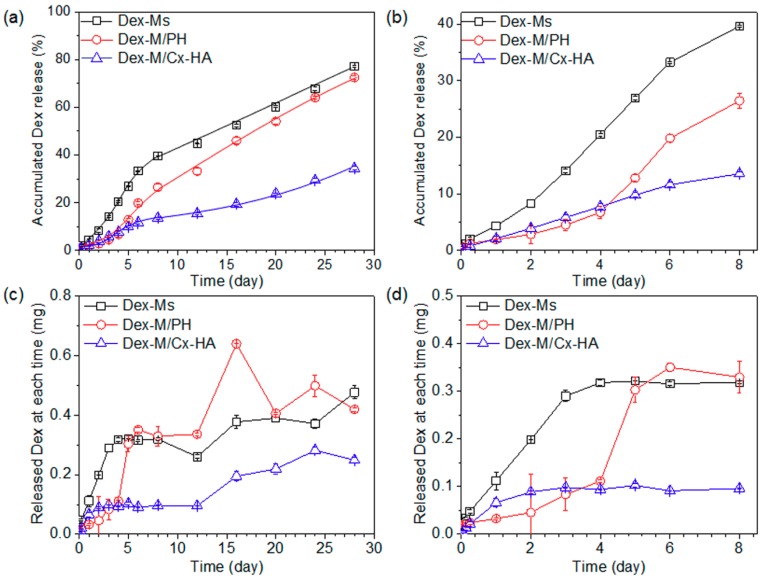
(**a**,**b**) Accumulated in vitro release of Dex and (**c**,**d**) the Dex percentage released by Dex-Ms alone, Dex-Ms/PH, and Dex-Ms/Cx-HA at each time point (**a**,**c**) for 28 days and (**b**,**d**) the enlarged period of eight days. (All data were performed three times and are presented as the mean and standard deviation).

**Figure 6 pharmaceutics-11-00438-f006:**
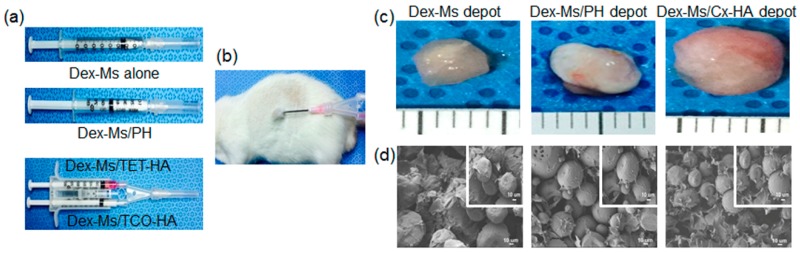
Images of the (**a**) injectable formulation of Dex-Ms alone, Dex-Ms/PH, and Dex-Ms/TET-HA and Dex-Ms/TCO-HA; (**b**) the Dex-Ms/Cx-HA depot formed in a Sprague Dawley rat after injection of Dex-Ms/TET-HA and Dex-Ms/TCO-HA; and (**c**) optical and (**d**) SEM images of the formed depots of Dex-Ms, Dex-Ms/PH, and Dex-Ms/Cx-HA after one week.

**Figure 7 pharmaceutics-11-00438-f007:**
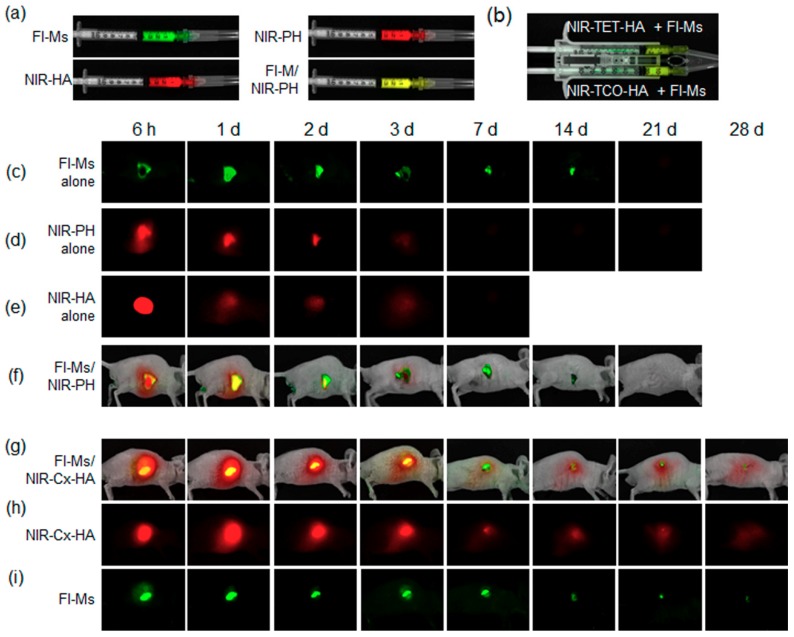
(**a**)NIR and fluorescent images for the single-barrel syringe formulation (FI-Ms alone, NIR-PH alone, NIR-HA alone, FI-Ms/NIR-PH) and (**b**) dual-barrel syringe formulation [(NIR-TET-HA + FI-Ms) and (NIR-TCO-HA + FI-Ms)]. In vivo fluorescent images of a nude mouse injected with a formulation of (**c**) FI-Ms alone, (**d**) NIR-PH alone, (**e**) NIR-HA alone, (**f**) FI-Ms/NIR-PH, (**g**) FI-M/NIR-Cx-HA, (**h**) NIR-Cx-HA (from FI-Ms/NIR-Cx-HA) and (**i**) FI-Ms (from FI-Ms/NIR-Cx-HA) for 28 days.

**Figure 8 pharmaceutics-11-00438-f008:**
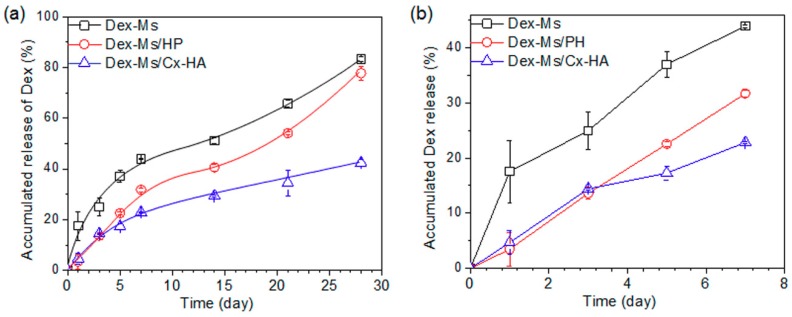
The accumulated in vivo release of Dex-Ms, Dex-Ms/PH, and Dex-Ms/Cx-HA for (**a**) 28 days and (**b**) the enlarged period of seven days. (All data were performed using three animals at each time point and are presented as the mean and standard deviation).

**Figure 9 pharmaceutics-11-00438-f009:**
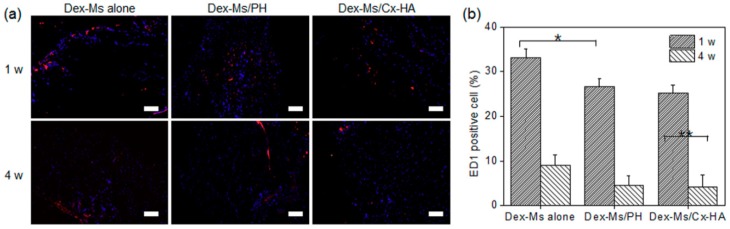
(**a**) ED1-stained histological sections of drug depot implants one and four weeks after implantation of Dex-Ms alone, Dex-Ms/PH, and Dex-Ms/HA in Sprague Dawley rats (scale bar: 100 µm) and (**b**) the percentage of ED1-positive cells in excised drug depot implants as a function of time after injection (* *p* < 0.05, ** *p* < 0.005). (All data were performed three times and are presented as the mean and standard deviation).

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
