# Peer review of "An Injectable Click-Crosslinked Hydrogel that Prolongs Dexamethasone Release from Dexamethasone-Loaded Microspheres"

_pharmaceutics, 2019, doi:10.3390/pharmaceutics11090438_

Round 1

Reviewer 1 Report

The paper entitled “An injectable click-crosslinked hydrogel that prolongs dexamethasone release from dexamethasone-loaded microspheres” describes the formulation of Dex-depot that could be embedded into pluronic (PH) and cross-linked hyaluronic acid (HA) gels and compares their properties as controlled drug delivery systems. The work plan has been well designed, the findings are clearly presented and nicely supported by experimental data.

However, I think that some minor revisions need being conducted:

·       Line 129: about freeze-drying procedure. A negative sign should be missing before 74 °C. A similar error was found in Line 271. Please, sort it out.

·       Lines 151-152: the sentence “potassium-containing phosphate-buffered saline” sounds odd since it sounds as metal potassium metal instead of potassium ions. On the other hand, I do not understand the relevance of being a “potassium-containing” buffer. Please, explain and rewrite.

·       Line 151: The authors used a sample of HA of 1.0 MDa. Is this correct?

·       Lines 150-156. Related to the preparation of TET-HA and TCO-HA, some questions should be answered:

o   How many mmols of carboxylic acid are content in the 100 mg of HA used? This is relevant since the TET and TCO groups will anchor in HA structure through this functional group.

o   Which were the degrees of functionalization with TET and TCO of -COOH groups that was aimed in HA?

o   Which were the degrees of functionalization achieved?

·       Line 178. In the NIR-labeled PH hydrogel formulation, why did the authors used maleic anhydride instead of, for example, succinic anhydride?

·       Lines 188-191. Why do the authors use a significantly different concentration of HA (2%) and PH (20%)? This should have a marked effect on the final properties of the formulations, such as the drug release.

·       Lines 210-211. The authors stated that “To measure the weight change of all formulations, the prepared PH, Dex-Ms/PH, Cx-HA or Dex-211 Ms/Cx-HA were individually added into a 15 mL vial,  …”. It is not clear whether the formulations were used as prepared in the previous section without any further treatment, such as freeze-drying, or they experienced a pre-treatment that it is not mentioned in the manuscript. Please, clarify it.

·       Lines 215-216. Related to the ratio of remained weight, the formula is not clear, since it is assumed that all measurements are conducted in dried samples, or just the final weight? Each term of the equation must be defined. On the other hand, and likewise mentioned before for Line 129, a negative sign seems to be missing. Please, sort out these points.

·       About in vitro and in vivo drug release.

o   Can the authors speak about in vitro drug release if the analyses have not been performed on an animal tissue or organ?

o   A comparison of the results from “in vitro” and in vivo analyses should be done.

·       Lines 230, 232, 234 and throughout the text. The expression “5 mg Dex concentration” is not accurate, since “mg” is not a concentration unit. Please, rewrite it.

·       Line 327. The authors stated that “The reactions showed a >90% yield of TET-HA and TCO-HA”. How did they calculate them?

·       Line 327. Details about how the NMR experiments were carried out should be included in the experimental part.

Author Response

Responses to the comments by Reviewer 1

We appreciate the reviewer’s comments. We have addressed each of these comments below and have highlighted revisions made to the relevant passages of the manuscript.

1) Line 129: about freeze-drying procedure. A negative sign should be missing before 74 °C. A similar error was found in Line 271. Please, sort it out

We added the missed sign in Lines 133 and 291.

2) Lines 151-152: the sentence “potassium-containing phosphate-buffered saline” sounds odd since it sounds as metal potassium metal instead of potassium ions. On the other hand, I do not understand the relevance of being a “potassium-containing” buffer. Please, explain and rewrite.

We revised the error in Line 155.

3) Line 151: The authors used a sample of HA of 1.0 MDa. Is this correct?

We used HA received from Humedix (Gyeonggi, Korea). We confirmed again the molecular weight of HA from Humedix company. We added the information in Line 120.

4) Lines 150-156. Related to the preparation of TET-HA and TCO-HA, some questions should be answered:

4-1) How many mmols of carboxylic acid are content in the 100 mg of HA used? This is relevant since the TET and TCO groups will anchor in HA structure through this functional group.

We described the explanation in Section 2.5 “100 mg of HA have 0.25 mmol of COOH group on HA. We designed 20% (0.05 mmol) of TET and TCO on HA”

4-2) Which were the degrees of functionalization with TET and TCO of -COOH groups that was aimed in HA?

In the previous work [reference 33], we compared the degrees of functionalization with TET and TCO for injectability and then used suitable degrees of TET and TCO (20% (0.05 mmol)) in this work.

4-2) Which were the degrees of functionalization achieved?

In this work, we designed the degrees of functionalization with TET and TCO for COOH groups on HA as 20%. We confirmed the functionalization by using elemental analysis and reported at the previous work [reference 33; NPG Asia Mater. 2019, 11, 30]. In the below table, TET-500 and TCO-500 means the 20% functionalization. The functionalization between calculation and founding in elemental analysis shows almost quantitative values. We described the result in Lines 343-344.

5) Line 178. In the NIR-labeled PH hydrogel formulation, why did the authors used maleic anhydride instead of, for example, succinic anhydride?

The reviewer’s point is well taken. We used maleic anhydride because we wish to determine ratio of functionalization of COOH through 1H-NMR measurement. So we described the explanation in Lines 196-197.

6) Lines 188-191. Why do the authors use a significantly different concentration of HA (2%) and PH (20%)? This should have a marked effect on the final properties of the formulations, such as the drug release.

HA solution has been usually prepared in 2 wt% solution because concentration above about 2 wt% showed high viscosity. While the prepared PH solution was prepared as a 20 wt% concentration, because PH has been widely used in 20 wt% as injectable in situ–forming hydrogel. Thus we described the explanation in Lines 185-187 and Lines 197-199 and added the related references [30,31]. Additionally, final properties of the formulation Figure 4c, d, e and f. Drug release behavior was examined and explained in Figures 5, 7 and 8.

7) Lines 210-211. The authors stated that “To measure the weight change of all formulations, the prepared PH, Dex-Ms/PH, Cx-HA or Dex-211 Ms/Cx-HA were individually added into a 15 mL vial, …”. It is not clear whether the formulations were used as prepared in the previous section without any further treatment, such as freeze-drying, or they experienced a pre-treatment that it is not mentioned in the manuscript. Please, clarify it.

We prepared the formulation of PH and Cx-HA as solution, and prepared Dex-Ms as powder form of microsphere. We already described in the section 2.8 and the added the explanation of formulation in Lines 232-234.

8) Lines 215-216. Related to the ratio of remained weight, the formula is not clear, since it is assumed that all measurements are conducted in dried samples, or just the final weight? Each term of the equation must be defined. On the other hand, and likewise mentioned before for Line 129, a negative sign seems to be missing. Please, sort out these points.

We determined the remained weight at predetermined time points. Thus we revised the explanation in Lines 227-230. The error of sign revised in Line 230.

9) About in vitro and in vivo drug release

9-1) Can the authors speak about in vitro drug release if the analyses have not been performed on an animal tissue or organ?

We performed the in vitro release from drug depot in vial, and in vivo release experiment using Sprague–Dawley rats. The in vitro and in vivo release of Dex described in experimental section 2.12 and 2.13, result section 3.3 and 3.5, and Figures 5 and 8.

9-2) A comparison of the results from “in vitro” and in vivo analyses should be done.

We already described the results and discussion between in vitro and in vivo analysis in each section and Lines 546-547 and 554-555.

10) Lines 230, 232, 234 and throughout the text. The expression “5 mg Dex concentration” is not accurate, since “mg” is not a concentration unit. Please, rewrite it.

The reviewer’s point is well taken. We revised the error as concentration unit thoroughly in manuscript.

11) Line 327. The authors stated that “The reactions showed a >90% yield of TET-HA and TCO-HA”. How did they calculate them?

We described the method about preparation and yield of TET-HA and TCO-HA in Lines 164-166.

12) Line 327. Details about how the NMR experiments were carried out should be included in the experimental part.

We added the explanation of NMR experiment in Lines 166-170.

We again appreciate the reviewer’s useful suggestions and comments. We have revised the manuscript to be in line with the reviewer’s comments as much as possible.

Reviewer 2 Report

This manuscript describes design and construction of microcapsules loaded cross-linkable hydrogels for in-vivo and in-vitro controlled drug release. 

Overall, the concept is good and the authors developed functional soft material. The methodologies are clear but the abbreviations used to describe the samples is somewhat confusing for a reader. As a result, reader has to refer to the experimental section to correlate with the results/data. 

This reviewer has following queries regarding the manuscript which needs to be addressed 

Microsphere formation is not clear what is the role of PVA solution? This should be clarified in the experimental or results and discussion section.  SEM images of microspheres - Are these spheres hollow - SEM images on the fragments of the sphere should provide this information. What is the thickness of the microsphere wall Are these microspheres porous - High resolution SEM images? Regarding in-vitro/in-vivo experiments - Do the microspheres disintegrate over the period of time?  What is the density threshold in which the microsphere-hydrogels soft materials can be obtained?  This reviewer appreciates the Rheology experiments but unfortunately the column plots does not provide changes in the profiles of storage and loss modulus of the microspheres incorporated within the hydrogels.  In-vivo - studies - the hydrogels are formed after injecting the functionalized HA but the key question is are these hydrogels able to disintegrate after being formed in the rats body?

Author Response

Responses to the comments of Reviewer 2

We are thankful for the reviewer’s careful analysis of our manuscript and subsequent suggestions. Firstly we added the abbreviations for samples and materials used in this work for a reader in Lines 586-589. We address each point below.

1) Microsphere formation is not clear what is the role of PVA solution? This should be clarified in the experimental or results and discussion section.

We appreciate the reviewer’s comment. We added the explanation of PVA role in the microsphere formation in Lines 127-129 and Lines 329-332.

2) SEM images of microspheres - Are these spheres hollow - SEM images on the fragments of the sphere should provide this information. What is the thickness of the microsphere wall Are these microspheres porous - High resolution SEM images?

In this work, we prepared microspheres not microcapsules of hollow form. Thus we added enlarged the cross-sectioned SEM image in Figure 2e’. We found the filled inner surface of microsphere not hollow. We added the explanation in Lines 332-333.

.

3) Regarding in-vitro/in-vivo experiments - Do the microspheres disintegrate over the period of time?

Generally PLGA degraded through in vivo hydrolytic chain scission of PLGA. The microsphere in this work gradually degraded for 4 weeks. We described the explanation in Lines 479-487.

Figure. Scanning electron microscope (SEM) images of Dex-M alone at 1, 2, and 4 weeks after injection

4) What is the density threshold in which the microsphere-hydrogels soft materials can be obtained?

In this work, hydrogels of HA or PH and formulation have considerable viscosity as shown in Figure 4. Thus the formulation of microsphere in hydrogels did not go down to the bottom by density for early period, although we did not investigate storage test long term.

5) This reviewer appreciates the Rheology experiments but unfortunately the column plots does not provide changes in the profiles of storage and loss modulus of the microspheres incorporated within the hydrogels.

We appreciate the reviewer’s comment. We showed the storage and loss modulus in the formulation in Figure 4d and e. We already described the explanation in Lines 401-403

6) In-vivo - studies - the hydrogels are formed after injecting the functionalized HA but the key question is are these hydrogels able to disintegrate after being formed in the rats body?

The reviewer’s point is well taken. We examined the in vivo persistence of the injectable materials by real-time imaging. From the Figure 7, PH and HA hydrogel disappeared after 3 days from subcutaneous tissue. Meanwhile, Cx-HA persisted in vivo for 4 weeks in Figure 7, and their signals gradually decreased due to the degradation of Cx-HA. Additionally we found the in vitro hydrogel persistence through maintain for 8 days and then gradually decreased by day 28. We described in Lines 408-411.

We again appreciate the reviewer’s kind suggestions and comments. We have revised the manuscript as much as possible to be in line with the reviewer’s comments.

Reviewer 3 Report

The in vivo sections of the work lack detail and raise some scientific and ethical concerns. For me, lines 227-229 are not enough of an ethical statement, particularly considering the number of time points you have chosen to use resulting in the euthanasia of maybe too many animals. Seven time points seems excessive and requires a better explanation of the logic behind this. 

How many rats were used in the study? You do not provide any n numbers and it's difficult to interpret the significance of the data. Please include n values in all Figures.

Other points to be addressed:-

Line 38 is poorly written and open to interpretation.  Delete "a" before drug release in line 40. Line 216 is missing info in the equation.

Author Response

Responses to the comments of Reviewer3

We are thankful for the reviewer’s careful analysis of our manuscript and subsequent suggestions. We address each point below.

The in vivo sections of the work lack detail and raise some scientific and ethical concerns. For me, lines 227-229 are not enough of an ethical statement, particularly considering the number of time points you have chosen to use resulting in the euthanasia of maybe too many animals. Seven time points seems excessive and requires a better explanation of the logic behind this.

We did that all experiments in this work were performed in accordance with the guidelines of the Institutional Animal Experiment Committee at Ajou University School of Medicine (approval No. 2016-0048). All animals used in this work were treated in accordance with the approved guidelines for the Care and Use of Animals for Experimental and Scientific Purposes. We added the explanation in Lines 241-244.

How many rats were used in the study? You do not provide any n numbers and it's difficult to interpret the significance of the data.

We did that the number of animals for in vivo release experiments added in the section 2.13. The detailed explanation added in Lines 254-256 and Lines 262-264.

Please include n values in all Figures.

We added performance time in all Figures.

Other points

Line 38 is poorly written and open to interpretation. Delete "a" before drug release in line 40. Line 216 is missing info in the equation.

The reviewer’s point is well taken. We revised the error.

We again appreciate the reviewer’s kind suggestions and comments. We have revised the manuscript as much as possible to be in line with the reviewer’s comments.

Round 2

Reviewer 3 Report

I accept your revision. 

I still have an issue with Line 39, it makes less sense now than previously! Are you trying to say that microspheres are capable of being loaded with several drugs simultaneously or that microshperes can be loaded with several different types of drug compounds? 

Author Response

Responses to the comments by Reviewer 1

We appreciate your and reviewer’s comments. We have revised Line 39 as much as possible in order to incorporate the suggestions from the reviewer.

We again appreciate the reviewer’s useful suggestions and comments.